# Network Bursts in 3D Neuron Clusters Cultured on Microcontact-Printed Substrates

**DOI:** 10.3390/mi14091703

**Published:** 2023-08-31

**Authors:** Qian Liang, Zhe Chen, Xie Chen, Qiang Huang, Tao Sun

**Affiliations:** 1Intelligent Robotics Institute, School of Mechatronical Engineering, Beijing Institute of Technology, Beijing 100081, China; liangqianbjpc@126.com (Q.L.); xiechen1116@163.com (X.C.); qhuang@bit.edu.cn (Q.H.); 2School of Medical Technology, Beijing Institute of Technology, Beijing 100081, China; zhechen95@bit.edu.cn

**Keywords:** 3D neuron clusters, microcontact printing, calcium dynamics, network bursts, pathological models

## Abstract

Microcontact printing (CP) is widely used to guide neurons to form 2D networks for neuroscience research. However, it is still difficult to establish 3D neuronal cultures on the CP substrate even though 3D neuronal structures are able to recapitulate critical aspects of native tissue. Here, we demonstrate that the reduced cell-substrate adhesion caused by the CP substrate could conveniently facilitate the aggregate formation of large-scale 3D neuron cluster networks. Furthermore, based on the quantitative analysis of the calcium activity of the resulting cluster networks, the effect of cell seeding density and local restriction of the CP substrate on network dynamics was investigated in detail. The results revealed that cell aggregation degree, rather than cell number, could take on the main role of the generation of synchronized network-wide calcium oscillation (network bursts) in the 3D neuron cluster networks. This finding may provide new insights for easy and cell-saving construction of in vitro 3D pathological models of epilepsy, and into deciphering the onset and evolution of network bursts in developmental nerve systems.

## 1. Introduction

With the advancement of biofabrication technologies, engineering neuronal networks in vitro to recapitulate the organization and functionality of neural circuits in vivo has attracted increasing attention in the last two decades [1,2]. It complements computational modeling and holds the potential to elucidate the functional mechanism of neural systems at both network and cellular levels, since it allows neuroscientists to custom-design neural circuits biologically [3,4,5,6,7]. Neuronal networks mimicking brain circuits can be built because dissociated primary neurons can form functional synapses even on non-physiological surfaces [8,9,10,11,12,13,14]. In the past, 2D-layered neuron networks were predominantly fabricated because of the advantages of easy operation and observation; however, 2D models may be confounded by atypical cell–cell or cell–matrix interactions and cellular morphology. To solve this problem, various 3D neuronal cultures have been developed, including reaggregate or sphere cultures, rotary bioreactor cultures with cell aggregates or microcarriers, hydrogel/scaffold cultures, and organotypic slice cultures [14,15,16,17]. 3D cultured neuronal structures have been shown to result in higher cell viability, longer neurite outgrowth, and different patterns of differentiation relative to 2D monolayers [18]. The 3D neuron cultivation technology is rapidly developing since it provides a biomimetic platform to elucidate the pathogenesis of neurological diseases and develop specific treatment drugs. However, due to the complex operation of cultivation techniques, the high cost of scaffold materials, and the limited capacity of nutrient supplements, it is still difficult to conveniently achieve large-scale 3D culture experiments [19,20].

Among the bio-interface technologies, microcontact printing (CP) has gained popularity in reconstructing neuronal networks in vitro, owing to its ease of use, high throughput, versatility, and reusability [19,20,21,22,23]. The CP substrate can produce a substantially thinner cell adhesion molecule (CAM) layer because of the limited adsorption time and the protein transfer process, compared with directly coating cell adhesion molecules [24]. Such a thinner CAM layer reduces cell–substrate adhesion compared with CAM-coated substrates. Neurons formed a homogeneous network on substrates with high cell-substrate adhesion, e.g., a carpet of glia [25] or a thick layer of coated CAM [26], and reaggregated to form a 3D clustered network on substrates with low cell-substrate adhesion, e.g., untreated glass substrates [27] or ploy-D lysine precoated polystyrene surface [28]. Compared with other 3D neuron cultivation technologies, the CP-producing neuron network was relatively large-scale because of sufficient nutrient supplement and simple culture procedure. Moreover, the thickness of 3D structures on the CP substrates was relatively thinner than other 3D cultures, which allows a clear fluorescent observation to study the network calcium dynamics [29]. Studies have shown that the neuron-substrate adhesion [27], seeding density [30,31], and local topological restriction [3,8,32] will impact not only the morphology and structure but also the spontaneous calcium dynamics of cultured neuronal networks, which reflects the intrinsic functional connectivity within certain networks [33]. However, on microcontact-printed substrates, the combined influence of these factors on 3D network structure and calcium dynamics remains to be investigated.

In this work, we explored the impact of reduced cell–substrate adhesion caused by CP substrate, seeding density, and local restriction on the aggregate formation of 3D neuronal constructs and their spontaneous calcium activity, as shown in Figure 1. We found that high 3D cell-aggregate degree induced by high cell-seeding density could be alternatively generated by relatively low seeding density with local restriction. The calcium dynamics of the resulting 3D structure showed the network-wide burst behavior of the aggregated neurons, which appears to play an important role in several aspects of the nervous system including development and integration in the sensory system but also in the initiation of pathological activity such as epileptic seizures [34].

## 2. Materials and Methods

### 2.1. Fabrication of PDMS Micro-Stamps

Stamps with two kinds of patterns were fabricated, including unpatterned stamps (with a uniform surface) and patterned stamps (with raised 2-by-1 modular micro-patterns) to fabricate Petri dish-wide neuronal networks and highly clustered networks at the microscale, respectively. First, chrome plates with a magnetron-sputtered chrome layer of 100 nm thickness and a pre-coated positive photoresist AZ4562 (AZ Electronic Materials Inc., Shanghai, China) with a thickness of 12 μm on soda-lime glass were purchased from Shaoguang Inc. (Beijing, China). Second, the chrome plate was cut into small square pieces of 25 mm width with a glazier’s diamond. Third, the small chrome piece was placed and fixed onto the vacuum holder platform of a Pattern Generator (microPG 101 from Heidelberg Inc., Heidelberg, Germany). Desired patterns were loaded and the direct laser writing process was initiated. Fourth, the exposed area of the photoresist layer was removed by developer AZ400K (AZ Electronic Materials, Luxembourg) 1:3 mixed with ultra-pure water (>18 MΩ*cm, Purist UV from Rephile Inc., Shanghai, China) for 7 min, and then the exposed area of the chrome layer was also removed by Chrome Etchant 18 (Dow Materials Science Inc., Midland, TX, USA) for 1 min. Each step was followed by rinsing with ultra-pure water three times. Fifth, a polydimethylsiloxane (PDMS) stamp was fabricated by replica molding. Before the developed and etched chrome plate was placed in a 35 mm Petri dish, two layers of aluminum foil were put in a 35 mm Petri dish, attaching closely to the bottom and wall, to facilitate the separation process. PDMS pre-polymer (Sylgard 184 from Dow Corning Inc., Midland, TX, USA) 10:1 mixed with curing agent was degassed at 5 °C for 30 min, and poured onto the Petri dish, followed by baking at 65 °C for 2 h. Finally, the cured PDMS stamp was released from the chrome plate and cut into the desired shape. The fabricated PDMS stamps were stored in ultra-pure water when not used to increase the hydrophilicity.

### 2.2. Fabrication of Surface-Modified Protein-Printed Substrates

First, the substrate of a 35 mm Tissue Culture-treated Petri dish (Dow Corning Inc.) was pre-coated by 0.2% agarose (BioFroxx Inc., Einhausen, Germany) solution (in ultra-pure water) to render the substrate cell-repellent. The pipetted solution should avoid touching the dish wall to render a relatively uniform thin layer of agarose gel. Second, protein ink [1% ECM gel (E1270 from Sigma-Aldrich Inc., Burlington, VT, USA) + 50 μg/mL poly-D lysine (PDL, P0899 from Sigma-Aldrich Inc.)] in PBS buffer solution (Thermo Fisher Scientific Inc., Waltham, MA, USA)] was prepared. The whole process should be guaranteed to be done aseptically. Filtering through a 220 nm filter is not allowed, because this would significantly reduce the protein concentration. Third, the stamp was pre-treated by coating a layer of sodium dodecyl sulfate (SDS, from Solarbio Inc., Beijing, China) to facilitate the protein transfer process. Before use, the PDMS stamp was put into a 10% SDS modification solution (in ultra-pure water). The solution was then sonicated for 5 min and left standing for 5 min [23]. After that, the stamp was taken out of the modification solution, dried with nitrogen, dipped into ultra-pure water to remove excess SDS, and dried with nitrogen again. Fourth, for the inking process, the modified stamp was placed inverted on a clean Petri dish. An amount of 200 μL PECM (pancreatic extracellular matrix) solution was pipetted onto the surface to cover the raised pattern area. The dish was covered and put in a 37 °C incubator for 20 min. Then, the stamp was taken out and dried by nitrogen quickly to avoid the formation of salt crystallization, which would greatly deteriorate the subsequent protein transfer process. Finally, in the printing process, the agarose-coated substrate was sterilized by ultra-violet radiation for an hour. Then, the stamp was once again inverted and pressed against the agarose-coated substrate, with the raised pattern against the substrate. A pressure of 300 g was applied for 2 min. After that, the stamp was released and the agarose-coated protein-printed substrate was obtained. The modified Petri dish was stored in a sterile clean bench. On the day before use, the Petri dish was supplemented with primary neuron culture solution overnight and observed the next day to rule out the possibility of bacterial infection.

### 2.3. Neuron Culture on CP-Based Substrates

Primary cortical neurons (OriCell SCCFN-00001 from Cyagen Inc., Beijing, China) from E18 Sprague Dawley rats were dissociated, cryopreserved in preservation medium (Neurobasal medium (Gibco Inc., Billings, MT, USA) + 10% DMSO), transported on dry ice, and preserved in liquid nitrogen before use. Right before plating, the cryopreserved primary neurons in 1 mL preservation medium were quickly thawed and supplemented with 4 mL pre-warmed neuronal plating medium (MEM + 5% fetal bovine serum + 5% horse serum + 0.6 D-glucose). Then, neurons were plated at the desired density on the agarose-modified PECM-printed substrates. The high-density and low-density seeding mentioned in the main text are 600 and 150 cells mm^2^, respectively. After 4 h, 2/3 medium was exchanged with neuronal culture medium (Neurobasal medium (Gibco) + 2% B27 + 1% GlutaMAX). Half of the medium was changed every 4 days from day in vitro (DIV) 4 with neuronal culture medium. No anti-mitosis drug was added to inhibit the proliferation of neuroglia cells because they are essential to the survival of the barely anchored neurons, and many neurons started to extend neurites only after the formation of glia carpet.

### 2.4. Calcium Imaging

Calcium imaging experiments were conducted on DIV 9, DIV 15, and DIV 22. In each experiment, cultured neurons were loaded with 4 μM calcium indicator Fluo-4 AM (Solarbio Inc.) and 0.01% Pluronic F-127 in Mg^2+^-free buffered saline solution (BSS, containing 130 mM NaCl, 5.4 mM KCl, 5.5 mM Glucose, 20 mM HEPES, and 1.8 mM CaCl_2_) at 37 °C for 40 min, rinsed with BSS, and incubated again in BSS for another 10 min before observation 2. The calcium indicator-loaded cultures were observed on an inverted microscope (IX73, Olympus Inc., Tokyo, Japan) equipped with a 10× objective lens, a short-arc lamp (X-Cite^®^ 120Q, Lumen Dynamics Inc., Ottawa, ON, Canada), and a charge-coupled device camera (DP21, Olympus). All calcium imaging videos were recorded by cellSens software at room temperature, with a temporal resolution of 80 ms (12.5 frames/s) and spatial resolution of 1200 × 800 pixels. The faintest level of excitation fluorescence was selected to reduce the adverse effects of photo-bleaching and phototoxicity.

### 2.5. Processing of Calcium Fluorescence Intensity Recording Videos

All recorded videos were processed preliminarily using ImageJ (version 1.53t, National Institutes of Health) to produce the time courses of absolute fluorescence intensity of manually selected regions of interest (ROIs). The recorded AVI files were imported and converted to grayscale image stacks to reduce the calculation complexity. Then, the brightness and contrast were adjusted to an appropriate level to make the boundary of neurons or clusters clear. After that, ROIs were selected as oval regions, and the fluorescence intensity traces were extracted, every point of which was the average intensity of a certain ROI on that frame.

The absolute fluorescence intensity traces of different ROIs were then fed into a custom MATLAB (version R2017b) script to output the relative intensity traces. First, the absolute fluorescence traces were filtered by a low-pass filter. Then, the intensity-diminishing effect caused by photobleaching and fluorescence dye leaking was compensated by first fitting with an exponentially decaying function and then being subtracted by this function. To avoid the intensity fluctuation caused by neuronal bursts, the traces were sampled as the lowest points by a window of 50 s width at 50 s stride. These lowest points were then fitted by an exponentially decaying function with a non-linear least square method. Next, the background fluorescence intensity *F_i_*_,0_ was calculated as the median of the 20% lowest intensity points of each compensated trace. Finally, the relative fluorescence intensity traces of each ROI *i* were calculated as *f_i_*(*t*) = (*F_i_* − *F_i_*_,0_)/*F_i_*_,0_ = Δ*F*/*F_i_*_,0_, with *F_i_* being the absolute intensity trace of ROI *i*.

### 2.6. Neuronal Burst Detection and Burst Trace Extraction

The onset and end time of each neuronal burst were inferred by a time-derivative thresholding method, which scanned the time derivative of the relative fluorescence traces. The onset time was labeled when the derivative reached the ±2.58 SD threshold from below, and the end time was labeled when the derivative reduced to zero from above after the onset time. The standard deviation (SD) for the derivative of each relative fluorescence trace was calculated after intercepting the derivative trace. The interception rule works as follows. First, the median of the derivative trace was determined. Then, a fixed-value range threshold (−0.1, +0.1) was offset by the median, and those not located in this range were eliminated. After that, SDs were calculated by the remaining derivative points. To reduce the false positive detection results, the onset times of detected neuronal bursts were double-checked by a five-frame derivative. More than that, those detected neuronal bursts with no apparent relative fluorescence elevation were erased by using a fixed-value threshold (median + 0.1). The median here was calculated as the median of the lowest 20% points of the relative fluorescence traces, rather than the one mentioned in the last paragraph.

### 2.7. Subnetwork and Network Burst Detection

A subnetwork was defined as the part of neurons (not all neurons in the whole network) that are connected with each other. In all cases, subnetwork burst was inferred to exist when at least *x* neuronal bursts with onset time within a certain time window *t_win_* were detected. The number threshold *x* was 2 for all three groups of networks. The time window *t_win_* was 240 ms (3 frames) for small-cluster large-scale networks and weakly-separated medium-scale networks, but was 400 ms (5 frames) for strongly-separated small-scale networks. The choice of time window was optimized to avoid the undesirable apparent division of one subnetwork burst into two separate ones. Considering the inevitable false-negative neuronal burst detection, any subnetwork burst in which at least 80% of all neurons in the network participate is inferred as a network-wide burst activity.

### 2.8. Correlation Coefficient

The correlation coefficient (CC) between neuron i and j, $r_{ij}$ was calculated using the respective relative fluorescence traces $f_{i}\left(t\right)$ and $f_{j}\left(t\right)$:$$r_{ij}=\frac{\sum_{t}\left(f_{i}\left(t\right)-\bar{f_{𝚤}}\right)\left(f_{j}\left(t\right)-\bar{f_{𝚥}}\right)}{\sqrt{\sum_{t}{\left(f_{i}\left(t\right)-\bar{f_{𝚤}}\right)}^{2}}\sqrt{\sum_{t}{\left(f_{j}\left(t\right)-\bar{f_{𝚥}}\right)}^{2}}}$$
where $\bar{f_{𝚤}}$ and $\bar{f_{𝚥}}$ are the average relative intensity of the two traces, respectively. For each network or subnetwork, the mean CC within the network or subnetwork was computed as:$$\bar{r}=\sum_{i\neq j}r_{ij}/N\left(N-1\right)$$
where N is the number of the network or subnetwork. The intra-subnetwork standard error of mean (SEM) was calculated as:$$SEM=\frac{\sqrt[{2}]{\sum_{i}\sum_{j}{\left(r_{ij}-\bar{r}\right)}^{2}}}{N\left(N-1\right)}$$
where i < j. The inter-subnetwork mean CC between two subnetworks was evaluated as $\bar{r}=\sum_{i}\sum_{j}r_{ij}/MN$, where i = 1, M, and j=M+1, N . M and N−M are the number of the two subnetworks, respectively. The inter-subnetwork SEM was calculated as:$$SEM=\frac{\sqrt{\sum_{i}\sum_{j}{\left(r_{ij}-\bar{r}\right)}^{2}}}{MN}$$

All statistical analyses were performed using Student’s unpaired *t*-test. ** Statistically significant difference with a *p*-value of <0.01, “ns” indicates no statistical difference.

### 2.9. Decay Time Constant

Every neuronal burst is followed by a decay phase, which represents the process of calcium ions being ejected from the cytoplasm. The speed of this process is quantified by the decay time constant, which was calculated by fitting the decay phase with an exponentially decaying function.

First, for each neuron, the decay phase after each neuronal burst was extracted and averaged. The overlapped decay phase of the relative fluorescence traces of a certain neuron is shown in Appendix A, in which the end time of the last neuronal burst corresponds to the onset of the decay phase, and the onset time of the next neuronal burst represents the end of this decay phase. Considering that the decay phase usually starts from a peak, which may lie inside the burst or shortly after the end time of the burst, the onset time of each decay phase is inferred as such a peak time. The peak was located by a local searching algorithm, which searched for the maximum in a time window ∆*t* followed by the end time of the last neuronal burst. The time window was locked to the next inter-burst interval *T_IBI_* by defining ∆*t* = *T_IBI_*/2 to balance computational efficiency and safety. The extracted traces were then averaged over time, with those time points containing only one trace being eliminated to reduce the noise.

Second, the averaged trace was fitted by an exponentially decaying function, which contains three unknown parameters: the initial state relative intensity *I_ini_*, the steady state intensity *I_ss_*, and the decay time constant *τ*. The function is:$$f\left(t\right)=\left(I_{ini}-I_{ss}\right)e^{-\frac{t}{\tau }}+I_{ss}$$

Third, the goodness of fit was calculated to evaluate the effect of fitting:$$R^{2}=1-\frac{\sum_{t}{\left(\tilde{f}\left(t\right)-\hat{f}\left(t\right)\right)}^{2}}{\sum_{t}{\left(\hat{f}\left(t\right)-\left(\sum_{t}\hat{f}\left(t\right)\right)/K\right)}^{2}}$$
where $\hat{f}\left(t\right)$ is the averaged trace, *K* is the number of time points of the averaged trace, and $\tilde{f}\left(t\right)$ is the time-discretized form of the fitted function. The average and SEM of *R*^2^ were evaluated for the decay time constant of each neuron in the corresponding network. The fitting was also evaluated and visualized by comparing the fitted trace and the averaged trace in Appendix A, where the good fitting effects were intuitively confirmed.

### 2.10. Burst Duration

Burst duration is defined as the duration of a certain neuronal burst, which was computed as the difference value between the onset and end time of the neuronal burst. The histograms of burst duration of all neuronal bursts are presented in Appendix A. The bin width is the minimum time resolution, i.e., 80 ms. Left and right limits are 0−*bin*/2 and max(*duration*) + *bin*/2, respectively, where max(*duration*) represents the maximum duration of all neuronal bursts. The distribution of burst duration was fitted by a Gaussian function:$$f\left(x\right)=\frac{1}{\sqrt{2\pi }\sigma }e^{-\frac{{\left(x-\mu \right)}^{2}}{2\sigma^{2}}}$$
where *µ* and *σ* are two unknown parameters to be ascertained.

### 2.11. Burst Frequency

Burst frequency represents the average number of neuronal bursts in one neuron within one second, which was calculated as the number of all neuronal bursts of a certain time window, divided by the product of the number of neurons and the duration of the time window. The width of the time window is set as 640 ms. The neuronal raster plot was scanned by the window at a stride of 80 ms to produce a time-dependent burst frequency vector. Average and SEM of burst frequency were calculated based on the vector to quantify the overall frequency and time-dependent fluctuation of burst activity, respectively.

### 2.12. Inter-Burst Interval (IBI)

IBI was computed as the difference between the onset time of two consecutive neuronal bursts. All IBIs computed in an experiment were grouped into a vector *V_IBI_*. Then, the bin width of the histogram was set as [max(*V_IBI_*) − min(*V_IBI_*)]/*C*, where the constant *C* was optimized to produce the maximum *R^2^* for Gaussian fitting considering the minimum temporal resolution, 80 ms. The left and right limits were 0 − *bin*/2 and max(*V_IBI_*) + *bin*/2, respectively. As in the burst duration analysis, the distribution of IBI was also fitted by a Gaussian function.

### 2.13. Coefficient of Variance of Inter-Burst Interval

The coefficient of variance of IBI (IBICV) for a certain neuron was calculated as the standard deviation (SD) of IBI distribution divided by the mean of IBI. IBICV distribution can reveal the activity pattern of a network. For the noise-free deterministic or rhythmic activity, IBICV equals zero. For random Poisson activity, IBICV distributes normally with a mean value near 1. In the simulations, 1000 independent neurons burst either rhythmically or randomly for 100 s at a mean burst frequency of 0.2 Hz. The time step was 0.1 ms.

## 3. Results and Discussion

### 3.1. Formation of 3D Aggregated Neuronal Structures on CP Substrates

First, we printed protein ink on an agarose-treated polystyrene substrate using an unpatterned polydimethylsiloxane (PDMS) stamp to explore how CP-reduced cell-substrate adhesion will influence the formation of neuronal network structure. We prepared three protein inks for neuron adhesion with the agarose substrates, including: 10% ECM gel + 50 μg/mL PDL in PBS (substrate 1), 1% ECM gel + 50 μg/mL PDL in PBS (substrate 2), and 50 μg/mL PDL in PBS (substrate 3), respectively. After a long time of cell culture, we found that neurons could fully spread and connect with each other to form a 2D-layered network on the substrate1 because of sufficient cell-adhesion sites, as shown in Figure 2a. For substrate 2, we found that neurons with high seeding density (~600 cells/mm^2^) tended to reaggregate into 3D neuron clusters, and each cluster consisted of 10–20 neurons. Subsequent inspection revealed that the synapses could enable the clusters to connect with neighboring ones, hence forming a single neuron cluster large-scale network, as shown in Figure 2b. Furthermore, we reduced the seeding density of neurons from 600 cells/mm^2^ to 150 cells/mm^2^, with other variables remaining unchanged. In this sparsely seeded un-patterned group, owing to the relatively long distance between neurons, neurons did not form clusters in the first few days after seeding. Neuron clusters started to form and spread neurites only after DIV 3, which may hypothetically be caused by the proliferation and spread of neuronal supporting cells, as they are known to be essential for the survival and maturation of neurons. Compared with the small clusters and their resulting network for high seeding density, the network for low seeding density could contain a number of neurons ranging from several to hundreds, and there were weakly separated networks with different clustering degrees, as shown in Figure 2c. For substrate 3, neurons self-aggregated and sporadically adhered to the surface without obvious growth behaviors because of a lack of cell adhesion sites, as shown in Figure 2d.

### 3.2. Calcium Dynamics in 3D Neuron Cluster Network

For all calcium recording videos, we first labeled all of the cells (including both neurons and non-neuronal cells) that varied in fluorescence intensity during the recording and calculated their decay time constant τ. Because neurons typically have a calcium decay time constant of less than 3 s [35,36], and the decay time constant of calcium transients of astrocytes reportedly ranged from 4 s to 10 s in response to ATP stimuli [37,38], those cells with τ > T_th_ (in this work, T_th_ = 4.0 s) were detected as non-neuronal cells and their calcium activities were therefore not considered in the follow-up analysis. Large-cluster [27,29] and homogeneous [39] networks with high cell numbers have demonstrated that synchronized calcium oscillation in the subnetwork or the whole network can be formed as early as DIV 5 after plating. A similar cell density has been used as the high cell seeding density for the fabrication of 3D neuron cluster networks on substrate 2. In the resulting networks, such calcium activity was not observed until DIV 15. On DIV 9, only sporadic cluster-level or sub-cluster bursts consisting of fewer than six neurons were observed under calcium fluorescence recording, as shown in Appendix A. Moreover, on DIV 15, a functional network formed, which was evidenced by spontaneous network-wide synchronized calcium oscillation (or “network bursts”) (Figure 3a) (see Appendix A). The network burst is a typical phenomenon in homogeneous cultured networks reported by other researchers [20]. On DIV 22, the synchronicity remained unchanged, as shown in Figure 3b. From DIV 15 to DIV 22, very few sporadic subnetwork or neuronal bursts were monitored in the observed area of the whole network, as shown in Figure 3c,d. On DIV 15, the correlation coefficient (CC) within the network was 0.9, which indicated the high functional connectivity within the network (Figure 3e). Figure 3f shows the mean CC slightly reduced from 0.90 to 0.88 on DIV 22, consistently with much research on homogeneous networks where network bursts remained dominant from their appearance [36].

Next, we developed an automated data processing framework to quantitatively analyze the burst waveform variation of the small-cluster network from DIV 15 to DIV 22. This framework calculated a number of important high-order characterizations by feeding the time-varying fluorescence traces of selected regions of interest (ROIs) (see Methods for details) [39]. As shown in Figure 4, from DIV 15 to DIV 22, in the network on substrate 2, the burst frequency increased from 0.15 Hz to 0.32 Hz, the burst duration reduced from 0.73 s to 0.38 s, the IBI was reduced from 6.68 s to 3.07 s, and the calcium decay time constant was also diminished from 2.01 s to 0.69 s, consistently with the intuitive observation. With network maturation, more synapses formed, the internal network information flow was reinforced, and a full-blown burst would be reached more quickly with stronger intra-network communication, i.e., fewer spikes induced in each neuronal burst, explaining the shorter burst duration [40]. Consequently, the inner peak calcium concentration was reduced, and the quiescent period (IBI) was shortened. Therefore, the calcium ejection constant was reduced and the burst frequency was increased. Moreover, a detailed analysis of the distribution of IBI in such networks on the two distinct time points revealed they could both be fitted well by Gaussian distributions, with R^2^ > 0.93 each time. It is highly possible that the network bursts happened rhythmically, added by a Poisson process. Further analysis of the distribution of IBICV confirmed such a hypothesis, consistently with other homogeneous networks reported [16,41]. This suggests that network bursts in homogeneous and small-cluster networks could share a similar initiation and propagation mechanism, in spite of their distinct network organization. The distribution of the burst duration in these two networks could also be fitted well by Gaussian distributions, with R^2^ both >0.98, revealing that the fluctuation of burst duration from its mean value was probably caused by random noise.

### 3.3. Calcium Dynamics of Networks Composed of Clusters with Obviously Distinguished Cell Number

To investigate the impact of diminishing inter-neuronal attraction on network organization, we reduced the seeding density of primary cortical neurons from the high seeding density of 600 cells/mm^2^ to the low seeding density of 150 cells/mm^2^, with other variables remaining unchanged. In this sparsely seeded unpatterned group, owing to the relatively long distance between neurons, neurons did not form clusters in the first few days after seeding. With the proliferation and spread of neuronal supporting cells, neurons sprouted neurites and built synaptic connections in the next few days. Gradually, many weakly separated networks with obviously different cell numbers formed. They were only weakly separated because the gap between different networks could potentially be bridged with the further maturation of the culture. Such networks contained a number of neurons ranging from several to hundreds and typically colonized an area smaller than 1 mm^2^. In Figure 5(a1), the two cluster networks with high cell numbers were so close to each other that they could not be morphologically grouped as two independent networks. However, calcium dynamics revealed that the two networks burst non-synchronously. In Figure 5(a2), two burst patterns appear and they switch between each other randomly, indicating that the neural signal is not relayed between the two networks. Further analysis reinforced this hypothesis by showing a significant distinction in CC between the two networks (Figure 5(a3,a4)), and higher R^2^ in separate networks compared to the merged one. Such a phenomenon of two neighboring networks bursting asynchronously was only reported previously for cortical networks on geographically constrained substrates [20,42,43,44].

More interestingly, in other networks with low cell numbers, time-varying subnetwork bursts appeared (Figure 5(b1)). The seven-neuron subnetwork was abstracted from a weakly separated network and treated as an independent network, in which abundant time-varying subnetwork burst patterns were observed. In the first 50 s, two network bursts and nine subnetwork bursts appear. Seven of the nine subnetwork bursts were distinct burst patterns. Compared with the 3D neuron cluster network in Figure 3, and the former two highly clustered networks in Figure 5a, where network bursts dominated the calcium dynamics, subnetwork patterns in this network were more complex. Self-organized cultured neuronal networks rich in subnetwork burst patterns, though beneficial for illuminating the evolution from sporadic neuronal bursts to network bursts, were rarely documented [20]. In a well-synchronized network, random neuronal bursts can induce the synchronized burst of other neurons, hence explaining the dominant network bursts. However, in this network, only neurons 3 and 4 burst synchronously, indicating the formation of a strong bi-directional synaptic connection, as shown in Figure 5(b2). No other neuron pair can promise synchronized calcium activity. The failed neuronal burst transmission may indicate an unreliable information communication between the neurons [45].

For the neuron cluster network in Figure 5a, there are higher cell numbers relative to the network in Figure 3, but network bursts could not be formed. We consider that low cell seeding density caused a low cell aggregation degree that could affect the generation of network bursts. It seems that networks with high aggregation degrees can induce network bursts because of the dense and more reliable inter-neuronal synapses, while less aggregated cell cluster networks could potentially present richer subnetwork activity patterns. To examine this hypothesis, we resorted to patterned µCP to confine neurons in desired areas and increase the aggregation degree. Meanwhile, other researchers have shown that symmetrically connected modular clustered networks also present dominant network bursts [20]. We wonder if asymmetry would break the reliable bidirectional signal transmission, hence allowing richer subnetwork bursts. To investigate this possibility at the same time, we changed the printing pattern to asymmetrically connected modular structures.

### 3.4. Calcium Dynamics of Neuron Cluster Networks in Local Restriction

We seeded primary neurons on agarose-treated PECM-patterned PS substrate with a low seeding density of 150 cells/mm^2^, with the patterns being asymmetrically connected modular structures. For the asymmetrical connection, the printed protein pattern is two squares connected by a connection area, which consists of a triangle (the width of it decreases progressively from left to right) and a thin line, as shown in Figure 6a. Such a pattern was designed to promote neurite growth from the left to the right area and inhibit the growth of reverse direction, therefore forming the asymmetrical neuronal network [46]. This pattern was arrayed 10-by-12 to yield 120 identical patterns. After culturing, we obtained both isolated networks (96/120) and asymmetrically connected modular networks (20/120) at one culture, all of which were highly clustered.

In the isolated networks, the hypothesis that stable network bursts could be induced by increasing the network’s aggregation degree was confirmed. Dominant network bursts reappeared in the highly clustered networks (Figure 6(b1)) (see Appendix A). More than that, other fascinating calcium behaviors were observed. First, contrary to previously reported observations [30] that network bursts could not be induced in small-scale networks, a nine-neuron network in our culture presented dominant network bursts (Figure 6(b2)). Second, repeated network bursts were observed in a network as small as a two-neuron couple (Figure 6c). Other research using a micro-pillar substrate reported a similar two-neuron synchronized bursting behavior [47]. However, the information flow in that study was unidirectional, not bidirectional. Specifically, if the two clusters (modules) of the network always presented simultaneous network bursts, then the network was considered bidirectional, while the network was considered unidirectional if calcium elevation was sometimes observed in only one cluster of the network (sub-network bursts). Moreover, an isolated neuron presented repeated spontaneous neuronal bursts. These repeated single-neuron bursts could not be induced by the diffusion of neurotransmitters from the nearby modules, because it paced asynchronously from them (Figure 6d). These three neurons may be intrinsically bursting neurons, which are prevalent in vivo and can present a burst of spikes by a short stimulation or by noise like miniature excitatory postsynaptic potentials [41,48,49].

In the asymmetrically connected modular networks, two subnetworks were located on two separate modules, which were connected by a nerve bundle. Owing to the printed area’s limited neuron-substrate adhesion and high inter-neuronal attraction, neurons tended to reaggregate into small clusters within modules. Under the attraction force of neurites between two clusters, they were attracted to each other, even outside the square area (Figure 6(e1)). However, in such an unusual circumstance, the network’s ability to transmit information was preserved, as evidenced by the repeated network bursts. Dominant network bursts were observed in all asymmetrically connected modular networks, as in symmetrically connected networks [20]. However, unlike in isolated, symmetrically connected, or cluster networks, where a short-term rapidly calcium-elevated phase was immediately followed by an exponentially decaying phase, in this asymmetrically connected modular network, an extended fluctuating plateau phase was inserted between the two phases (Figure 6(e2)). This could have been caused by the reverberatory signal transmission of neuronal bursts between modules [19]. The extended fluctuation plateau phase could possibly be explained by the reverberatory signal transmission between the two modules of the network. In particular, the plateau phase of the calcium transient represents reverberating neuronal bursts within the network so that the calcium concentration can be maintained at a high level. According to this hypothesis, for a neuron that just produces a neuronal burst, before substantial calcium decays and after the refractory period, another neuronal burst signal is just transmitted back to trigger another neuronal burst, therefore maintaining the calcium concentration for a long time (the extended fluctuation plateau phase). Moreover, the duration of such a plateau phase between two consecutive network bursts greatly varies, indicating the variability of small-scale networks in calcium dynamics, consistent with other research [20].

## 4. Conclusions

In this paper, we proposed a CP-based culture method to conveniently construct a 3D neuron cluster network on a 2D surface. The effect of the construction condition including adhesion between neuron and substrate, cell seeding density, and locally restricted patterns on the network functions was carefully analyzed according to the calcium dynamics of the networks. Furthermore, we found that synchronized network-wide calcium oscillation (network bursts) can be generated in the network with low cell number and high cell aggregation. When epilepsy occurs, neurons in the brain are often abnormally excited and discharged at the same time due to a certain stimulus. Our proposed CP-based network can reproduce such behavior in vitro; therefore, CP-based 3D neuron cultures may provide a potential platform for the incident mechanism study of epilepsy and the responding drug testing.

## Figures and Tables

**Figure 1 micromachines-14-01703-f001:**
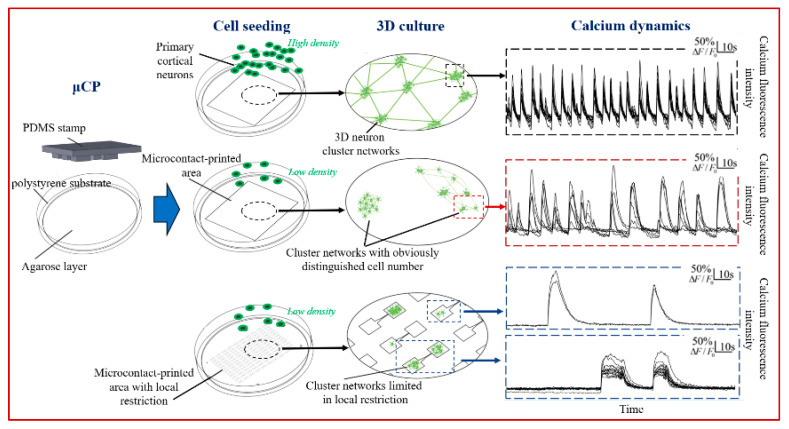
Schematic diagram of 3D neuron networks with different cell-aggregation structures formed on microcontact-printed substrates by varying printed patterns and seeding density, and presenting organization-related calcium activities. Top panels: with a uniform pattern, a Petri dish-wide network composed of 3D neuron cluster networks forms, with calcium activities characterized by network bursts. Middle panels: by seeding neurons with low cell density, networks composed of various 3D neuron clusters with different cell numbers form, among which the one with fewer cells presents time-varying subnetwork bursts. Bottom panels: by confining neurons with the same low cell-seeding density in small squares, we obtain highly clustered networks, where dominant network bursts reappear. F indicates fluorescence intensity.

**Figure 2 micromachines-14-01703-f002:**
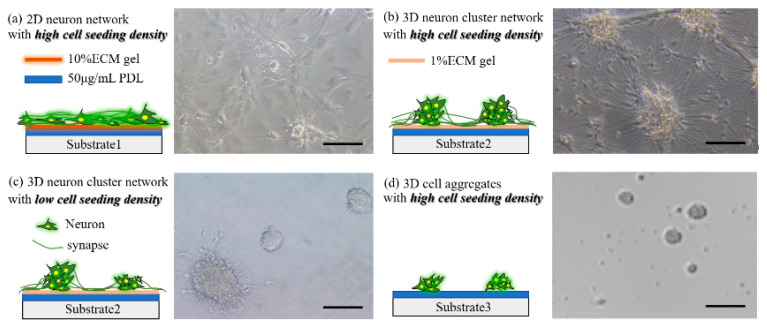
3D aggregated neuronal structures. (**a**) Neuron network monolayer on substrate 1; (**b**) 3D neuron cluster networks and their resulting large-scale network on substrate 2; (**c**) 3D neuron cluster networks with obviously distinguished cell number on substrate 2; (**d**) 3D neuron aggregates on substrate 3. All scale bars are 50 µm.

**Figure 3 micromachines-14-01703-f003:**
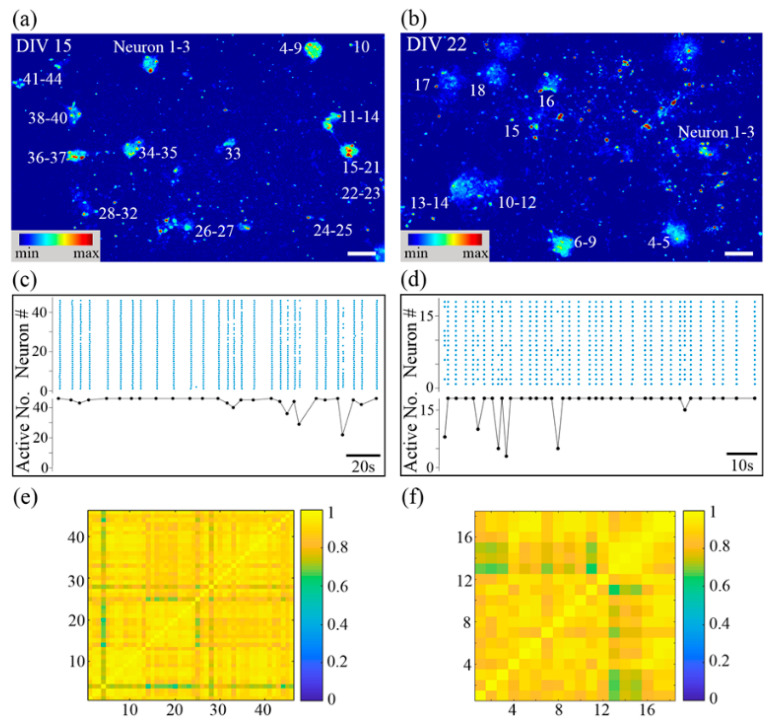
Calcium dynamics of 3D neuron cluster network with high cell seeding density on DIV 15 and DIV 22 are both dominated by synchronized network bursts. False-color fluorescence micrographs of the network on DIV 15 (**a**) and on DIV 22 (**b**). (**c**,**d**) Represent DIV 15 and DIV 22, respectively. Top panel: raster plot of neuronal bursts; bottom panel: number of active neurons in one subnetwork burst versus time. The matrix of CC on DIV 15 (**e**) and on DIV 22 (**f**). Scale bar: 100 µm.

**Figure 4 micromachines-14-01703-f004:**
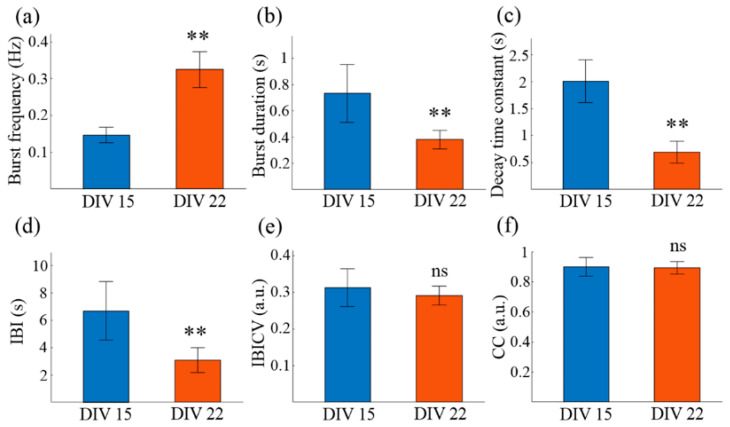
Characterization and comparison of the calcium dynamics of the 3D neuron cluster network on DIV 15 and DIV 22. (**a**) Neuronal burst frequency, the average number of neuronal bursts per neuron per second. (**b**) Neuronal burst duration, the difference value between the onset and the end time of a neuronal burst. (**c**) The fitted decay time constant of the calcium attenuation phase. (**d**) Inter-burst interval (IBI), the interval between two consecutive neuronal bursts. (**e**) Coefficient of variance of IBI (IBICV), the standard deviation of the IBI distribution across neurons divided by its mean value. (**f**) Correlation coefficient (CC). ** *p* < 0.01, ns indicates no statistical difference. Data are presented with mean ± SEM.

**Figure 5 micromachines-14-01703-f005:**
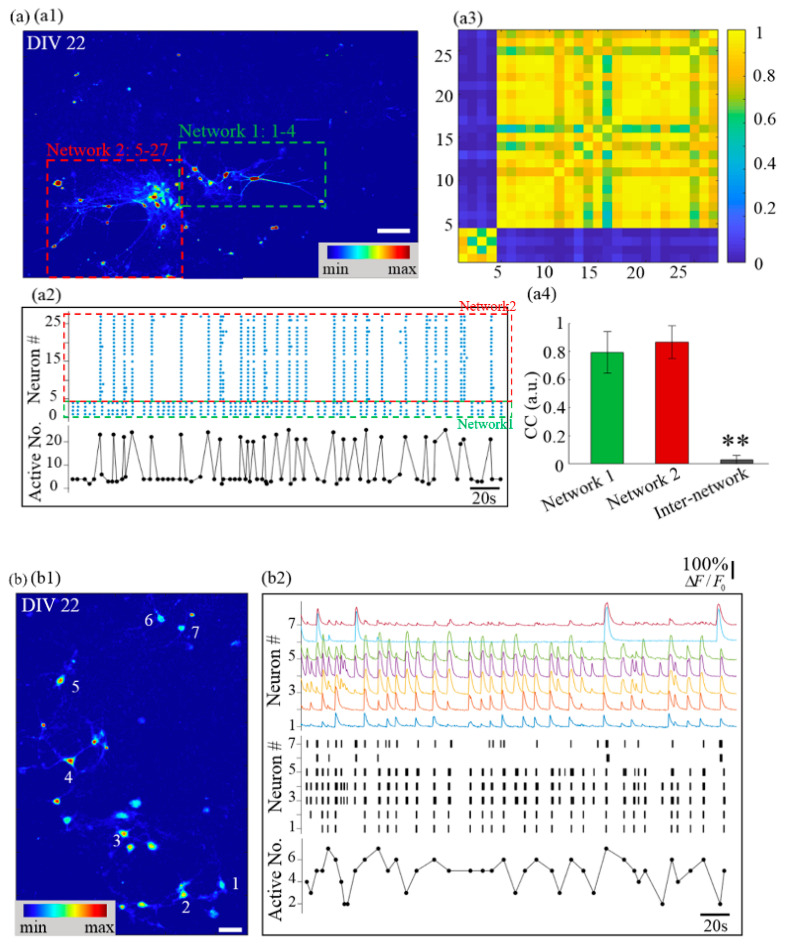
Networks with different cell number present different calcium dynamics, ranging from dominant network bursts to time-varying subnetwork bursts. (**a**) Two neighboring neuron networks with high cell number. (**a1**) False-color fluorescence micrograph of two neighboring highly-clustered networks. (**a2**) Top panel: raster plot of neuronal bursts, bottom panel: number of active neurons. (**a3**) Matrix of CC. (**a4**) CC of intra- and inter-network. (**b**) Neighboring networks with low cell number. (**b1**) False-color fluorescence micrograph of a less dense network consisting of seven neurons, with their locations indicated. (**b2**) Top panel: relative fluorescence traces of the seven neurons in (**b1**); middle panel: the black lines represent location of neuronal burst, with the line width coded by burst duration. Bottom panels: active neuron number in a subnetwork burst versus time. Data are presented with mean SD. ** *p* < 0.01 against other two networks. Scale bar: (**a1**) 100 μm, (**b1**) 40 μm.

**Figure 6 micromachines-14-01703-f006:**
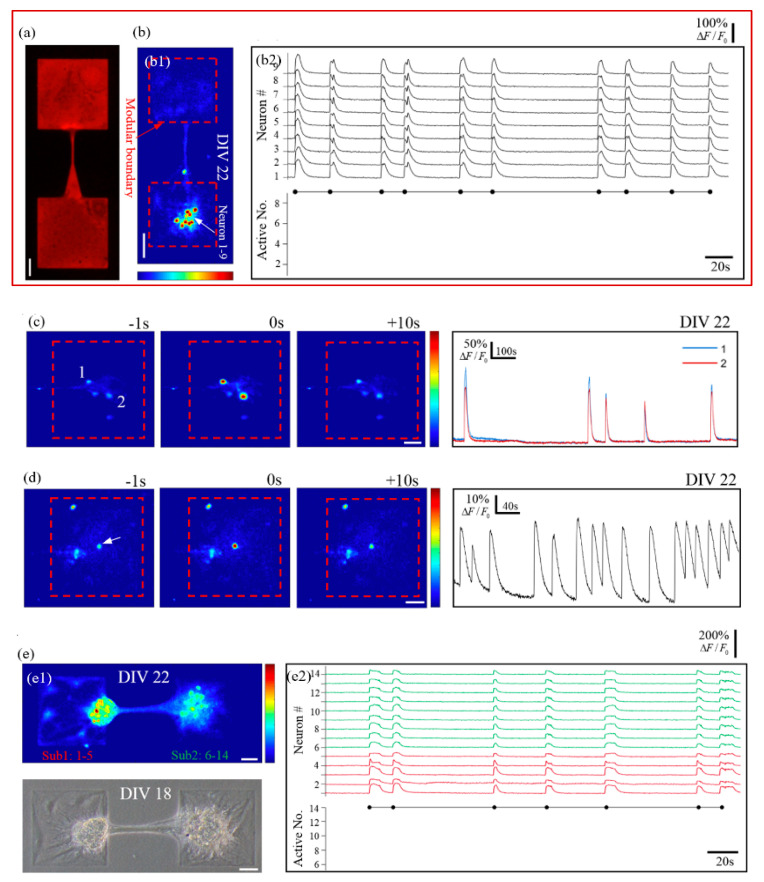
The locally-confined highly-clustered networks show dominant network bursts. (**a**) Asymmetrical modular pattern. (**b**) An isolated multi-neuron network. (**b1**) False color micrograph of an isolated network. (**b2**) Top panel: relative fluorescence traces of neurons from (**b1**); bottom panel: active neuron number in a subnetwork burst versus time. (**c**,**d**) Are from two isolated networks consisting of only two and one active neuron(s), respectively. Left three panels: False-color calcium recordings at a second before, right on, and ten seconds after a peak time, respectively; rightmost panel: relative fluorescence traces of selected neurons (loci indicated in the leftmost panels). (**e**) An asymmetrically-connected modular network. (**e1**) False-color (top panel) and phase-contrast (bottom panel) micrographs of the modular network. (**e2**) Top panel: color-coded traces of neurons in two subnetworks present synchronized neuronal bursts with post-burst plateau phases; bottom panel: active neuron number in a subnetwork burst versus time. All scale bars: 50 μm.

## Data Availability

The data presented in this study are available on request from the corresponding author.

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
