# Peer review of "Network Bursts in 3D Neuron Clusters Cultured on Microcontact-Printed Substrates"

_micromachines, 2023, doi:10.3390/mi14091703_

Round 1

Reviewer 1 Report

This paper reports a nice set of experiments to characterize structural and activity aspects of neural networks, which falls within the scope of this journal. Under the opinion of this reviewer, several aspects must be revised in this paper:

-          Several statements of the introduction must be supported by references: lines 41, 44, 46, 59, and 61, after the period at the end of the sentences in all the cases.

-          The last paragraph of the introduction is actually discussing results, and it should be restructured and shortened.

-          For the general audience, it would be handy that the introduction described and cited the basics of the neuronal bursts, and the role of calcium dynamics in this process.

-          The motif of the patterns fabricated should be described already in section 2.1.

-          The hydrophilicity increase stated in lines 113-114, and the facilitated protein transfer stated in lines 124-125, should be supported by references or by experimental results.

-          Line 130, the meaning of PECM should be defined.

-          Fix typos along the paper, for example lines 39, 127, 161, 254, 338, etc.

-          In section 2.7, define the meaning of “subnetwork”.

-          It is not clear how many replicates and images of each condition were coonsidered in sections 3.1, 3.2, and 3.3. It should be defined, and it should be big enough to support the conclusions stated in the paper.

-          Fix the units in the scale bars in most figure captions.

-          The content of the paragraph in lines 500-510 is out of the scope of section 3.4. It should be removed from this section, and omitted or integrated with the conclusions.

-          In general, the text in Results and discussion include extensive descriptions of the data already presented in the figures, which becomes redundant.

-          The sentence “Our proposed CP-based network … can provide a potential platform for the incident mechanism study of epilepsy and the responding drug testing …”, is not supported enough by the results of this work to be claimed by the authors. It could be commented as potential idea for future experiments, but not as a direct conclusion from these results. To avoid misleading conceptions for the readers, the comment on epilepsy should be removed from the abstract and from the introduction, and Figure 7 should be removed or moved to supplementary information.

Fix typos

Reviewer 2 Report

The authors describe the formation of calcium bursts in 3D neuron networks cultured on microcontact-printed substrate. In particular, they investigated how cell-substrate interactions, the cell density, and the topological organization of the neuronal ensembles influence the formation of either network or subnetworks and their burst activity. 

The topic is of general interest for scientists attempting to mimic in vitro the physiological and pathophysiologocal features of neuronal network activity. However, the manuscript raises several major concerns that must be addressed before further consideration for publication: 

1.  For calcium imaging, the calcium indicator Fluo-4 AM has been used. As this dye can equally enter any cell type and it enables to detect calcium activity of not only neurons but also of astrocytes and neural stem cells, how can the authors be sure that they are assessing the calcium activity of neurons rather than astroglial cells? The latter ones are also present in the culture as clearly stated by the authors themselves. It will be important to prove that the analysis has been carried out specifically on neurons, by either using a neuron-specific fluorescent calcium biosensor as the genetically encoded GCaMP or, in alternative, by using red-fluorescent neurons for Fluo4AM studies. 

2. Astrocytes are mentioned to be essential for the formation of synapses and neuron network functional development, especially regarding the low density cultures which develop activity later compared to the high density cultures (lines 321-325). However, this statement is not supported by data, i.e. immunostaining of astrocytes and of synapses at the time points selected for the calcium imaging study would be relevant.

3.  Authors report the number of neurons in each cluster.  However, it is unclear how did they count the neurons. Did they do by optical microscopy or by fluorescence microscopy of Fluo4-AM labelled cells? In both cases, it is not possible to consider those cells as neurons. On the contrary a neuron-specific biomarker, e.g. MAP2, must be used to provide an accurate count of the neurons.

4. Calcium activity is reported already at 5 days and then at 9 days after seeding (lines 339-342) but no data are shown to demonstrate the presence of bursts at these early time points. In addition, these are two very early time points in which the activity derives likely from glial cells rather than from neurons. So, using neuron-specific dyes would be important to understand the network evolution. 

5. The schematic in figure 2 shows 4 different types of neuron cultures. However, the first one, i.e. high density culture on the very hydrophilic substrate 1 promoting high cell-substrate interaction, is not described in terms of calcium activity. It is fundamental to provide data on the network and subnetwork activity present in such a type of standard culture, to fully understand the relevance of the different models proposed by the authors. 

6. The author mention the difference about their asymmetrically connected modular structures and the symmetric modular clusters reported already in the literature. However, it is unclear how they obtained the asymmetric distribution and how their pattern can be considered asymmetric at all. This should be clarified with a specific schematic accompanied by better description. It is also unclear how they can consider their clusters bidirectional and distinguish them from unidirectional clusters reported in literature.  

7. In lines 492-496, the authors ascribe the extended fluctuation plateau phase of their modular networks to the asymmetric connection. This is not understandable. They should better justify their observation. Also, the main reason could be that in their case there is a connection of just 2 modules while in the former literature where a fast fluorescence decay is reported, there are 4 interconnected modules. This should be more deeply discussed.

8. The authors make a parallelism between the highly syncronized network activity observed in their high density confined cultures and epilepsy. This looks like a bit of a stretch, as epileptic seizures are not simply the results of highly synchronous networks but they rather follow specific patterns. 

Therefore here, the authors should use at least a pair of well-recognized anti-epileptic drugs or prepare low density cultures of neurons isolated from epileptic animal models to demonstrate that the activity observed in the high density and confined cultures of wild-type neurons resembles epileptic activity.  

The manuscript presents in some parts the inappropriate use of prepositions such as "on" that makes difficult to understand the meaning of the sentences, e.g. lines 15-18 in the abstract; or it changes the meaning, e.g. line 131. In some lines (e.g. 61, 353, 416) the punctuation is incorrect. Some words are inappropriate, e.g. downregulated referring to IBI (line 368). In addition, there are many typos throughout the manuscript. It should be carefully revised.

Reviewer 3 Report

In this manuscript, the authors have demonstrated an “3D” culture technology of neurons. There are several points the authors need to address to improve the manuscript.

1. The layouts of the images, e.g. Fig. 1, are very messy and difficult to read.

2. Micro-contact printing (CP) technology is widely used to form 2D networks. But the authors claimed that their the “cell clusters” in the culture dish were “3D”. It is well known that it was easy to form “cell clusters” even with any intervention, like CP or “3D” restriction. The authors should clarify the novelty of their study.

3. In vitro models of epilepsy is an interesting topic. However, no strong evidence was presented in this manuscript to support the authors’ claim.

Round 2

Reviewer 2 Report

The authors have only partially addressed the concerns raised during the 1st round of revision.

In particular, they did not report the calcium activity of 2D homogenous neuron cultures. While it is true that these have been extensively studied, here they represent an internal control needed for comparison to the  3D neuron clusters. In addition, as the authors show the image of such 2D cultures in figure 2 side by side to the other types of cultures, it is expected that, along with that, they would show the analysis of calcium activity of all the presented conditions. 

The authors claim to have recorded the activity at DIV 9 and to have observed "only sporadic cluster-level or sub-cluster bursts consisting of fewer than 6 neurons" at this early time point, however they display only DIV 15 and 22 in figure 3 and 4. The claim must be supported by data and figure.

The number of days in cultures is reported as DIV or day. They should choose one of the two, ideally DIV for freshly isolated cells, and report it consistently throughout the manuscript.

Authors have not addressed the language concerns raised before.  

Please check the comments of the 1st round of revision. They have not been addressed.

Reviewer 3 Report

The revised version could be accepted.

Author Response

Thank you

Round 3

Reviewer 2 Report

The authors have addressed the raised concerns and the manuscript can be accepted in the present form.